# Microbiota-Derived Natural Products Targeting Cancer Stem Cells: Inside the Gut Pharma Factory

**DOI:** 10.3390/ijms24054997

**Published:** 2023-03-05

**Authors:** Valentina Artusa, Luana Calabrone, Lorenzo Mortara, Francesco Peri, Antonino Bruno

**Affiliations:** 1Laboratory of Innate Immunity, Unit of Molecular Pathology, Biochemistry and Immunology, IRCCS MultiMedica, 20138 Milan, Italy; 2Immunology and General Pathology Laboratory, Department of Biotechnology and Life Sciences, University of Insubria, 21100 Varese, Italy; 3Department of Biotechnology and Biosciences, University of Milano-Bicocca, 20126 Milan, Italy

**Keywords:** cancer stem cells (CSCs), drug resistance, gut microbiota, microbiota-derived metabolites, bioactive compounds, natural products

## Abstract

Cancer stem cells (CSCs) have drawn much attention as important tumour-initiating cells that may also be crucial for recurrence after chemotherapy. Although the activity of CSCs in various forms of cancer is complex and yet to be fully elucidated, opportunities for therapies targeting CSCs exist. CSCs are molecularly distinct from bulk tumour cells, so they can be targeted by exploiting their signature molecular pathways. Inhibiting stemness has the potential to reduce the risk posed by CSCs by limiting or eliminating their capacity for tumorigenesis, proliferation, metastasis, and recurrence. Here, we briefly described the role of CSCs in tumour biology, the mechanisms involved in CSC therapy resistance, and the role of the gut microbiota in cancer development and treatment, to then review and discuss the current advances in the discovery of microbiota-derived natural compounds targeting CSCs. Collectively, our overview suggests that dietary intervention, toward the production of those identified microbial metabolites capable of suppressing CSC properties, is a promising approach to support standard chemotherapy.

## 1. Introduction

Nowadays, several highly successful cancer therapies are available, with the majority of regimens combining surgery, radiotherapy, and medicine, which includes chemotherapy, targeted therapy [1], and most recently, immunotherapy [2]. The type and stage of the cancer being treated determine which techniques should be employed. One of the most important goals in cancer biology is to discover cells and signalling pathways that are essential for tumour regression, thus developing novel drugs that can abrogate the growth and metastasis of malignant tumours. Among medications, conventional cancer chemotherapy remains one of the most widely used approaches. Traditional chemotherapy is an aggressive form of cytotoxic drug therapy that destroys all rapidly proliferating cells, whether they are malignant or not. Thus, this method also destroys perfectly healthy cells. On the contrary, mechanism-based therapies, such as targeted therapy and immunotherapy, are designed to find and slow the growth of cells that possess a specific cancerous phenotype. Compared to the scatter-gun approach of chemotherapy, targeted therapy appears more sniper-like, accurately destroying its target without causing any collateral harm to otherwise healthy cells. Because targeted therapies only target cancer cells, some patients report fewer side effects than those with chemotherapy, which in turn presents many bottlenecks, including a lack of specificity, which has an impact on healthy tissues, as anticipated, but also rapid drug metabolism and both intrinsic and acquired drug resistance, all contributing to decreased efficacy [3,4]. In this scenario, understanding the molecular mechanisms of cancer and tumour cell biology represents an area of investigation that poses a unique challenge to clinical oncologists and cancer researchers. Here, after introducing CSCs and their role in cancer biology, we briefly describe the mechanisms involved in CSC therapy resistance. Next, we focus our attention on the gut microbiota and its relationship with cancer development and treatment. The main purpose of our review is to provide a comprehensive summary of the currently available literature describing microbiota-derived natural compounds targeting CSCs.

## 2. Role of Cancer Stem Cells in Tumour Cell Biology

CSCs describe a class of stem-like cells of tumour origin that behave similarly to normal stem cells in their ability to regulate their cell cycle by switching between a quiescent and a differentiation state. This includes key stem cell features, such as self-renewal [5] and the capability to differentiate into parental tumour cells. Moreover, CSCs participate in fundamental processes of tumour growth and progression, including cancer cell proliferation, metastatic spread, and immune evasion. According to the literature, CSCs exist in most haematological and solid tumours. A cluster of differentiation (CD)133+ CSC population was revealed in colorectal cancer (CRC) in 2007 [6,7] after CSCs were first identified in 1994 in acute myeloid leukaemia (AML) [8]. Since then, their significance in solid cancer has been thoroughly researched. To date, the advent of modern flow cytometry and cell sorting techniques has allowed for the identification of cell populations with CSC features, based on their expression of specific markers. Indeed, human CSCs were recognised in other solid tumours, including breast [9], brain [10], prostate [11,12], lung [13], and pancreatic [14,15] tumours. Notably, in non-obese diabetic/severe combined immunodeficient (NOD/SCID) mice, as few as 100 CSCs were sufficient to produce tumours [9]. Nowadays, CSCs are identified and classified according to the markers they express, including cell surface antigens, stemness-related markers (OCT4, SOX2, and NANOG), or high aldehyde dehydrogenase (ALDH) activity. To complicate the picture, CSC surface marker expression varies by tissue type and even by tumour subtype. For example, CD44^+^CD24^−/low^ and ALDH^+^ CSCs were characterised in breast cancer [16,17], along with CD133^+^CD44^+^ in colon [18,19], brain [20], and lung [21] cancer; CD34^+^CD8^−^ in leukaemia [22]; CD44^+^ in head and neck tumours [23]; CD90^+^ in liver cancer [24]; and CD44^+^/CD24^+^/ESA^+^ in pancreatic cancer [25]. CSCs were at first thought to make up only a small portion of a solid tumour’s overall cell population; however, according to some estimates, up to 25% of cancer cells may display CSC characteristics [26]. Regarding the genesis of CSCs, a variety of theories have been proposed. According to one theory, CSCs develop from healthy stem/progenitor cells when they undergo a specific genetic mutation or environmental change that confers to them the capacity to cause tumours. In terms of cellular characteristics, phenotype, activity, and also cell surface markers, certain CSCs exhibit similarities to typical stem/progenitor cells, thus lending credence to this notion [27]. A second explanation describing the origin of CSCs contends that they originate from healthy somatic cells that undergo genetic and/or heterotypic changes to develop stem-like properties and malignant behaviour. Emerging data showing that CSCs are resistant to standard chemotherapy and radiation treatment and are very likely to be the cause of cancer recurrence and metastasis have enhanced the clinical significance of CSCs [5,28,29].

## 3. Therapy-Resistant Nature of Cancer Stem Cells

Chemoresistance, recurrence, and metastasis remain the primary causes of cancer mortality, advances in therapeutic development notwithstanding. Numerous investigations have revealed that a small subgroup of cancer cells, called CSCs, is the cause of the tumour’s recurrence. Some regulatory signalling pathways, including the Wnt/β-catenin, Sonic Hedgehog (SHH), and Notch pathways, which are important in the self-renewal process, are shared by CSCs and regular stem cells [30]. Accumulating evidence has shown that the expression of markers related to stemness is crucial for tumour maintenance and that these molecules also mediate cancer therapy resistance. Furthermore, resistant CSCs might cause metastasis at a distant site, resulting in the formation of a metastatic tumour [31]. The mechanisms through which CSCs adapt to escape cancer therapy are summarised in Figure 1 and further discussed below.

### 3.1. Cell Cycle Arrest and Quiescence

Strong proof of a connection among CSCs, tumour cell plasticity, cell-cycle quiescence, and immune suppression in cancer originates from a wide range of publications. Several studies have shown that CSCs can conceal themselves from the immune system at the onset, avoiding detection during the immunosurveillance phase. Cell cycle is a multi-phased, intricate, and tightly regulated process. Cell cycle control requires the phase-specific transcription of cell cycle genes. Mutations in cell cycle genes can make healthy cells more inclined to acquire a malignant phenotype [32]. In a very elegant study, Agudo et al. [33]. demonstrated that fast-cycling cells, such as Lgr5+ stem cells detected in the stomach, ovaries, and mammary glands, experienced immune clearance. Conversely, slow-cycling stem cells, such as those in muscle and hair follicles, were resistant to just EGFP death-inducing (Jedi) T-cell eradication. Furthermore, the ability of latent stem cells to autonomously downregulate the antigen-presentation pathway via the transactivator NLRC5 is crucial for immunological escape. Notably, the process is reversible once stem cells enter the cell cycle [34]. It has been speculated that cancer cells use the characteristics of dormant stem cells to evade immune cell identification (Figure 1A). In this regard, it was recently shown that CSCs have immune-evasive properties when they enter quiescence [35]. Accordingly, in xenotransplant investigations, leukaemia CSCs were discovered to be chemotherapy-resistant and to be in the G0 (resting) phase of the cell cycle [36]. We can therefore envisage that the immunologically privileged status of CSCs is dependent on their capacity to adopt a quiescent state. Indeed, CSCs’ pharmacological resistance results from a mismatch between their relatively slow cell cycle [37] and the rapidly proliferating cancer cells that multiple chemotherapeutic treatments are designed to target.

### 3.2. Autophagy

Organelles, protein aggregates, and intracellular pathogens are the types of cellular cargo that are engulfed by double-membraned vesicles called autophagosomes during the evolutionarily conserved catabolic process known as autophagy, which results in their destruction and recycling after fusion with the lysosome [38]. CSCs exhibit autophagy reliance equal to that in tissue-resident stem cells (Figure 1B). For example, it was demonstrated that the secretion of interleukin (IL)-6 from CD44^+^/CD24^low/−^ breast cancer cells is dependent on autophagy and necessary for CSC maintenance [39]. In addition, autophagy is induced by a wide range of cancer therapies. For example, Imatinib™, a small molecule tyrosine kinase inhibitor used to treat metastatic gastrointestinal stromal tumour (GIST), causes the induction of autophagy in GIST cells [40]. According to preclinical data, stress-induced autophagy helps CSCs survive, while blocking autophagy can help in overcoming CSC resistance [41]. In the case of Imatinib™-treated GIST cells, tumour cell apoptosis was induced by inhibiting autophagy, using the lysosomotropic drug chloroquine (CQ) [40]. Moreover, in prostate cancer, clomipramine (CMI), CQ, or metformin treatment enhanced apoptosis and dramatically reduced cell viability by blocking autophagy in enzalutamide-resistant cells, overcoming the resistance to enzalutamide, an inhibitor of the androgen receptor signalling pathway used for the treatment of metastatic castration-resistant prostate cancer [42].

### 3.3. Tumour Microenvironment

As per normal stem cells, CSCs are frequently found in anatomically separate locations, hidden niches within the tumour microenvironment (TME) that provide a protective physical and chemical environment from direct contact with drugs and the host immune system. In tumour niches, intricate interactions between cells and the extracellular matrix (ECM) create a complex environment that determines stem cell resilience and the preservation of stemness. ECM remodelling also impacts CSC survival (Figure 1C). On one hand, a physical barrier created by enhanced ECM stiffness can protect CSCs from chemotherapeutic drugs. On the other hand, ECM degradation by matrix metalloproteinases (MMPs) can allow for the release of cytokines and growth factors that enhance tumour cell invasion, metastasis, and angiogenesis [43]. Moreover, solid tumours are commonly affected by hypoxia. The capacity of the pre-existing blood vessels to meet the oxygen requirement is frequently exceeded in cases of uncontrolled cell multiplication [44]. When under hypoxic and therapeutic stress, CSCs use a variety of signalling pathways that are modulated by hypoxia-inducible factor (HIF) signalling to modulate their stemness. HIF-induced gene products include epithelial-to-mesenchymal transition (EMT) programmers, glycolysis-associated molecules, drug resistance-associated molecules, miRNAs, and VEGF [45]. Therefore, by maintaining CSCs in their undifferentiated stem cell state, which enables self-renewal and the accumulation of epigenetic and genetic mutations, hypoxic environments may promote the formation of malignant clones [46]. In addition, the TME has been shown to have an acidic extracellular pH, which is a consequence of lactate accumulation via increased anaerobic glycolysis in hypoxic conditions [47]. In that respect, it was recently demonstrated that extracellular acidosis may cause cancer cells to develop stem-like characteristics and aid in the proliferation of the CSC subpopulation [48]. Lastly, tumour cells, inflammatory cells, cancer-associated fibroblasts, and CSCs are just a few of the cell types that belong to the specialised microenvironment known as the perivascular niche, which is found right next to blood vessels. Here, the stemness features of CSCs, such as their capacity for self-renewal, multipotency, and tumorigenic potential, are maintained by molecular interactions among various cell types [49].

### 3.4. Drug Inactivation

CSC chemoresistance has also been linked to intracellular drug inactivation (Figure 1D). A class of detoxifying enzymes known as ALDHs is frequently upregulated in cancer cells leading to treatment resistance. ALDHs are overexpressed in cancer cell clusters with stem-like characteristics, where they contribute to the defence of cancer cells by converting harmful aldehydes into more soluble and less reactive carboxylic acids [50]. For example, ALDH is crucial for contrasting the effects of diverse chemotherapeutic agents, such as cyclophosphamide, irinotecan, temozolomide, paclitaxel, doxorubicin (DOX), and epirubicin [51,52,53,54,55]. In addition, ALDH has been a widely used marker for CSC identification. Increased metabolic activity, along with conventional anticancer drugs, leads to aldehyde generation, which results in DNA double-strand breaks (DSBs) via reactive oxygen species (ROS) and lipid peroxidation. Thus, the overexpression of ALDH is essential for CSC survival. Moreover, it can inhibit immunogenic cell death (ICD) and cause the activation and growth of immunosuppressive regulatory T cells (Tregs), thus influencing immune cell activity in the TME [50]. Additionally, in NOD/SCID mice, acute myeloid leukemic cells that possess increased ALDH activity seem to have more capacity for engraftment compared to their ALDH-negative counterparts [56]. Moreover, the epigenetic inhibition of thymidine phosphorylase has been observed in CSCs, resulting in the therapeutically inefficient transformation of active 5-fluorouracil (5-FU) and methotrexate [55,57,58]. Finally, CSCs use thiol glutathione to inactivate platinum [59].

### 3.5. Drug Extrusion

One of the primary defence mechanisms for CSCs is the transcription of multifunctional efflux transporters from the ATP-binding cassette (ABC) gene family (Figure 1E) [60]. By using the energy of ATP hydrolysis to adenosine diphosphate (ADP) [61], these transporters actively efflux peptides, inorganic anions, amino acids, polysaccharides, proteins, vitamins, and metallic ions [62]. Intrinsic CSC-chemoresistance has been associated with their ability to express proteins of the family of ABC transporters, which results in drug extrusion and loss of effectiveness. Increased ABC transporter expression, including ABCB1 (P-glycoprotein/MDR1), ABCC1 (MRP1), and ABCG2 (BCRP), is one of the most well-established strategies for cancer cells to acquire multidrug resistance (MDR) [63]. A plethora of drugs that modulate MDR-ABC transporters have been developed during the past years, and some of them have also demonstrated significant efficacy in clinical trials [63]. However, one must bear in mind that in addition to promoting the growth of tumours, stem cell-driven tissue repopulation also promotes the growth of adult-specific normal tissues, such as the bone marrow, digestive tract, and hair follicles; thus, the complete inhibition of ABC transporters could have severe drawbacks.

### 3.6. Altered DNA Damage Response (DDR)

A large number of chemotherapy treatments, including platinum-based drugs and radiation, kill cancer cells by causing DNA damage. Studies have demonstrated that CSCs are incredibly effective in repairing DNA damage (Figure 1F) [64]. CSCs’ resistance to DNA-damaging therapies is thought to be caused by this enhanced DNA damage response (DDR). DDR is an extremely intricate network made up of numerous pathways, each of which exhibits cross-talk both within the network and with other signalling pathways [65]. When compared to non-stem tumour cells, CSCs have a higher capability for DNA repair either through increased DNA repair pathways or through delayed cell-cycle progression [66]. The MRE11–RAD50–NBS1 (MRN) protein complex, a major sensor of DNA double-strand breaks, is expressed in both normal and cancerous cells, as well as CSCs. However, the MRN function is improved in CSCs through interactions with the CSC-related molecules Notch1, ALDH1A1, CD44, SHH, and BMI1, in contrast to that in non-stem tumour cells [67], or through CD171, which boosts CSCs’ radioresistance and selectively triggers the DNA damage checkpoint [68]. The resting activation status of checkpoint kinases could serve as a crucial defence mechanism for CSCs against genotoxic chemicals when coupled with the induction of DNA repair. Not unexpectedly, several DDR-inhibitory drugs are currently undergoing pre-clinical and clinical testing [66]. In addition, stem cells regulate self-renewal and differentiation via differential configurations of the chromatin structure; thus, it is expected that histone changes and chromatin remodelling following DNA damage differ between stem cells and developed cells. In recent years, it has been evident that chromatin’s epigenetic dysregulation plays a significant role in CSCs development and frequently plays a crucial part in CSCs’ self-renewal throughout tumour growth [69].

### 3.7. Epithelial-to-Mesenchymal Transition (EMT)

Several fundamental features of cellular physiology undergo modifications as a result of the epithelial-to-mesenchymal transition (EMT) program, including alterations to cell morphology, which are related to changes in the cytoskeletal organisation; the dissolution of epithelial cell-cell junctions; loss of apical-basal polarity and concomitant gain of front-rear polarity; acquisition of the ability to breakdown and reorganise the ECM, thus enhancing motility and allowing cell invasion; and alterations to the expression patterns of at least 400 different genes [70]. The relationship between the EMT program and the CSC state raises the possibility that non-CSCs can become CSCs by enacting this program (Figure 1G) [71,72]. Indeed, EMT has been also linked to chemoresistance [73,74]. Worthy of note, an EMT-associated gene-expression signature has been strongly linked with treatment resistance, based on examinations of the relationships between the clinical outcomes of individuals and the gene-expression profiles of the associated tumour samples [75,76]. Moreover, by activating the EMT program, cancer cells can form metastatic colonies [74,77]. More specifically, according to recent studies, cells undergoing partial EMT may exhibit hybrid E/M phenotypes, possess more stem cell-like features, and exhibit more resistance to drugs than cells undergoing complete EMT. Additionally, partial EMT facilitates collective cell movement as clusters of circulating tumour cells or emboli, enhancing cancer cells’ capacity for metastasis and tumour genesis at the secondary regions [78].

### 3.8. Vasculogenic Mimicry

There is a unanimous understanding that solid tumours require a sufficient blood supply to grow. The term vasculogenic mimicry (VM), first coined by Maniotis [79], describes the ability of aggressive cancer cells to form de novo perfusable, matrix-rich, vasculogenic-like networks in a way that differs from traditional tumour angiogenesis in that it does not rely on endothelial cells. These new patterns of tumour microcirculation assist in perfusing rapidly growing tumours, removing fluid from leaky arteries, and/or integrating with the body’s endothelial-lined normal vessels [80]. The link between VM and poor clinical outcomes in patient malignancies suggests that VM confers a survival advantage to the aggressive tumour cell phenotype [81,82]. Additionally, preclinical pharmacological studies have shown that VM is connected to anticancer therapy resistance [83]. A significant amount of data suggests that CSCs aid in the development of VM (Figure 1H) [84]. The VM phenotype of tumour cells has a molecular signature that includes upregulated expression of genes related to embryonic progenitors, endothelial cells, vessel formation, matrix remodelling, and coagulation inhibitors, as well as downregulated expression of genes primarily related to lineage-specific phenotype markers [80,85].

### 3.9. Acquisition of Stemness Due to Treatment

It has been shown that chemotherapy and radiation both foster CSC traits in non-stem cancer cells and might even cause non-stem cancer cells to become CSCs [86,87] (Figure 1I); thus, the issue of CSCs not responding to conventional cancer treatments goes beyond the simple inability of these treatments to eradicate CSCs. The plasticity of cancer cells enables the transient acquisition of stemness-related traits. After receiving carboplatin treatment, hepatocellular carcinoma cells developed stem-like characteristics, including the ability to self-renew and the expression of stemness-related genes (*SOX2* and *OCT3/4*), which demonstrated the potential for chemotherapy to generate stemness [88]. Moreover, after being exposed to the chemotherapeutic drug 5-FU, human gastric cancer cell lines demonstrated resistance to 5-FU, as well as characteristics of stemness, such as tumorigenicity and the ability to self-renew [89].

Despite chemotherapy substantially eliminating a large portion of the tumour volume, there cannot be a noticeable clinical improvement if CSCs have not been eradicated to provide long-term disease-free survival. Therefore, CSCs are thought to be a significant target for the development of new anticancer drugs, being that CSC-focused therapy is a key driver for any effective anticancer strategy. In addition to synthetic drugs targeting CSC pathways (reviewed in [30]), dietary components, mostly (poly)phenolic compounds, have shown the ability to inhibit tumour progression [90] and angiogenesis [91]. Nearly all of these naturally occurring phytochemicals with chemopreventive activities also have antioxidant and anti-inflammatory effects. Interestingly, several mechanisms involved in the anticancer effects of dietary phytochemicals target pathways involved in CSC stemness maintenance [92]. Of note, human-ingested nutrients can be transformed by the gut microbiota into useful microbial compounds that closely link diet to cancer [93]. Indeed, the microbiota-derived metabolome has the potential to encourage or prevent carcinogenesis in organs distant from the gut. An emerging field in anticancer research examines the intricate interactions between particular gut microbial metabolites and the advancement or inhibition of cancer cell proliferation [94].

## 4. Role of the Gut Microbiota in Cancer

The gut microbiota comprises a multitude of microorganisms, mainly bacteria across over 500 species, of which the number reaches 10^13^–10^14^, similar to the number of cells in an adult human [95,96]. The majority of them (about 90%) is represented by two bacterial phyla, the Gram-positive *Firmicutes* (*Bacillus* spp., *Lactobacillus* spp., and *Clostridium* spp.) and the Gram-negative *Bacteroidetes* (*Bacteroides* spp. and *Prevotella* spp.) [97,98]. In their entirety, gut bacteria have several functions, including food fermentation, vitamin production, protection against pathogens, and immune response stimulation; thus, the intestinal microbial balance is highly relevant to human health [99]. It has been established that the breakdown of the host’s and gut microbiota’s symbiotic relationship can facilitate the onset of numerous disorders, including autoimmune disease [100,101] and cancer [102]. In this scenario, the molecular basis of various long-established epidemiological relationships between certain bacteria and cancer are presently being studied [103]. For instance, the correlation between *Helicobacter pylori* and the risk of the development and progression of gastric cancer, but also the case of *Fusobacterium nucleatum*, of which the role in the setting of CRC has been extensively studied [104,105,106,107,108,109,110]. Bacterial infections were associated with cancer stemness in both cases. In the former case, Bessède et al. observed that following *H. pylori* infection, gastric epithelial cells overexpressed CD44 and acquired CSC features, while in the latter case, Cavallucci et al. revealed that *F. nucleatum* can contribute to the microbiota-driven colorectal carcinogenesis by directly stimulating colorectal CSCs [111,112]. Additionally, Ha and colleagues provide evidence that EMT and cancer stemness acquisition are induced in oral cancer cells by prolonged infection with *Porphyromonas gingivalis* [113].

Moreover, there have been documented indirect effects of the gut microbiota on the growth of tumours in tissues outside of the gastrointestinal tract [110]. It is fascinating to note that the gut microbiota, by releasing bacterial products that can enter the bloodstream, can practically influence all host organs and systems and eventually affect cancer progression. This expanding knowledge points out that intestinal dysbiosis may cause carcinogenesis in localised gastric and intestinal cancers and tumours located in distant regions of the body [103,110]. For instance, lipopolysaccharide (LPS), a component of the Gram-negative bacterial cell wall, which is recognised by Toll-like Receptor 4 (TLR4), is one of the molecules derived from gut bacteria that has been demonstrated to promote cancer [110]. In a model of chronic injury-induced liver cancer, LPS-induced TLR4 stimulation increased the expression of the hepatomitogen epiregulin in stellate cells, which had a pro-tumorigenic effect [114]. Additionally, deoxycholic acid (DCA), a metabolite produced by gut bacteria, has also been linked to an increased risk of developing hepatocellular carcinoma when its level is increased due to dietary- or hereditary obesity-induced shifts in the gut microbiota composition [115].

On the other side of the coin, recent studies have observed that the gut microbiota can also exert immunomodulatory and anti-tumoral effects in cancers. For instance, in a rat model, the probiotic bacteria *Lactobacillus acidophilus* have been found to decrease the occurrence of CRC [116]. Moreover, exopolysaccharides from *Lactobacillus* spp. were able to slow down cell division in a time-dependent fashion and trigger apoptosis by upregulating the expression of Bax and caspase 3 and 9, while downregulating Bcl-2 and survivin, in a colon cancer cell line (HT-29) [117]. Abdelghani et al. provided a comprehensive list of anti-cancer compounds derived from microbial metabolism and their anticancer activities, which range from apoptotic, anti-proliferative, and cytotoxic activity to chemosensitisation to 5-FU [118].

Along with the investigation of the links between the gut microbiota and cancer, the microbiota of tumours themselves has received some consideration. Interestingly, more research into the microbiota revealed that it was also present within tumour tissues that were previously assumed to be sterile [119]. Furthermore, the local microenvironment and the tumour immunological context seem to interact with the tumour-associated microbiota, or microbial communities found in the tumour or inside its body compartment, ultimately affecting cancer growth and the response to therapy [120]. The intratumoral microbial community further complicates the cancer–microbiota–immune axis, which significantly impacts T-cell-mediated killing and anti-tumour immune surveillance [121]. Recently, Zhou et al. reviewed the hitherto neglected but significant impacts of the small molecules derived from tumour microbiota metabolism on the TME and their essential roles in cancer development [122]. Not only that, numerous instances of the microbiota altering drug metabolism and interfering with immunotherapy have been reported [123,124,125,126], and it is expected that research in this area will continue.

From the perspective of “therapeutic microbiology”, the host’s health status can be improved through a variety of approaches: (a) by introducing living, beneficial bacteria (known as probiotics), influencing the microbial composition (probiotics) [127]; (b) providing non-digestible substances, such oligofructose, oligosaccharides, inulin, raffinose, and stachyose (known as prebiotics), which are fermented by endogenous colonised probiotics in the large intestine (colon), promoting the establishment of beneficial microbiota [128]; (c) administering microbial metabolites with low molecular weights (<50, 50–100, and <100 kDa) that have positive effects on health (postbiotics) [129,130]. A significant number of published studies that discuss the capability of postbiotics to regulate different cellular processes and metabolic pathways have been published in the literature and reviewed elsewhere [130,131]. However, the microbiota remains an untapped avenue for finding small-molecule drugs for cancer treatment.

## 5. Microbiota-Derived Metabolites with Activity towards CSCs

Diet and environmental exposures, as well as lifestyle, have a major role in influencing the human gut microbiota composition and its metabolic activity, which can have an impact on health [132,133,134]. CSCs are very dependent on their surroundings for their energy supply; thus, nutrients play a pivotal role in modulating CSC growth or stemness. Over the past few decades, numerous studies have attempted to clarify the processes governing CSCs’ response to diet [135]. The anaerobic microbial population ferments undigested dietary components and host products, primarily mucin, to produce a remarkably wide range of metabolites that reflect both the chemical diversity of the dietary substrates and the microbiota’s unique metabolism [136]. As outlined above, microbiota metabolites, defined as intermediate end products of microbial metabolism, are key players in the microbiota–cancer relationship. These metabolites can be categorised based on two different parameters: origin (intracellular or extracellular) and function (primary or secondary), respectively. While secondary metabolites are produced close to the stationary phase of growth and are not essential for growth, reproduction, or development, primary metabolites are required for the optimal growth of bacteria. Several studies were conducted to assess the health-promoting effects of microbial products; in those cases, researchers described them as ‘biogenic’, ‘cell-free supernatant’, ‘abiotic’, ‘metabiotic’, ‘paraprobiotic’, ‘ghost probiotics’, ‘pseudoprobiotic’, ‘supernatant’, etc. [137]. Only in 2013, the term “postbiotics” was created to describe soluble components secreted by living bacteria or released following bacterial lysis, including enzymes, peptides, teichoic acids, muropeptides derived from peptidoglycan, polysaccharides, cell surface proteins, and organic acids [129]. This definition also gained support from further reports [138,139]. A detailed and exhaustive description of the range of metabolites produced by gut microbial metabolic activity and their roles in health and diseases is beyond the scope of this review and can be found elsewhere [140]. Here, we focus exclusively on the documented effects of microbiota-derived metabolites that specifically target CSCs and their features.

Traditional approaches to identifying novel bioactive natural products include extraction, fractionation or isolation, chemical characterisation, and, ultimately, an assessment of the potential beneficial effect through the execution of biological assays [141]. In this connection, cell-free supernatant (CFS), a solution that contains metabolites produced as a result of microbial growth, represents an invaluable metabolite-rich source. For instance, the antioxidant, antimicrobial, and anticancer properties of CFS have been demonstrated [142,143,144]. In 2016, An and Ha showed that the expression of particular CSC markers, CD44, CD133, CD166, and ALDH1, can be inhibited by *Lactobacillus plantarum* (LP) supernatant. Besides that, combined treatment with LP supernatant and 5-FU: (1) prevented CRCs from surviving and caused cell death by inducing caspase-3 activity; (2) prompted an antitumor mechanism by inactivating the Wnt/β-catenin signalling pathway in chemoresistant CRC cells; and (3) decreased the formation and volume of colonospheres [145]. Later in 2020, the same authors also demonstrated that in 5-FU-resistant CRC cells (HCT-116/5FUR), *Lactobacillus plantarum*-derived metabolites (LDMs) boost drug sensitivity and have antimetastatic effects as well. By reducing the expression of claudin-1 (CLDN-1), co-treatment of HCT-116/5FUR with LDMs and 5-FU decreased chemoresistance and metastatic activity. Their findings suggested that targeting 5-FU-resistant cells with LDMs and 5-FU cotreatments can be effective [146]. Moreover, Maghsood et al. treated human colon cancer stem-like cells enriched from an E-cadherin shRNA-engineered HT-29 cell line (HT29-ShE) with size-fractionated *Lactobacillus reuteri* CFS. Their results showed that crude and >50 kDa fractions of CFS significantly decreased the expression of COX-2, a crucial factor in the maintenance and function of CSCs. In addition, they demonstrated that colon cancer stem-like cell apoptosis and cell proliferation were both suppressed by *L. reuteri* CFS [147].

Diet plays a major role in cancer aetiology and prevention; thus, a healthy diet can be a game-changer factor [148,149,150,151,152]. Moreover, food is a significant source of substrates for the production of microbial metabolites. Amongst the vastness of microbiota-derived metabolites, some have been identified as potential CSC-targeting molecules (Figure 2).

### 5.1. Butyrate

Non-digestible carbohydrates, including resistant starch, non-starch polysaccharides, and certain soluble oligosaccharides, reach the large intestine without undergoing any digestion, because of the upper intestine tract lacks certain food-digesting enzymes [153,154]. Short-chain fatty acids (SCFAs) and gases are produced through the anaerobic degradation of such non-digestible fibres by gut microorganisms. SCFAs are aliphatic carbon-based acids, with acetate (C2), propionate (C3), and butyrate (C4) being the most abundant [155]. Several studies have found a link between a high-fibre diet and a lower risk of colon cancer [156,157,158]; this drove scientists toward the investigation of SCFA’s role in carcinogenesis prevention. However, when studying butyrate, researchers faced a contradictory effect: if butyrate effectively inhibited the proliferation of undifferentiated, highly proliferative adenocarcinoma cells while promoting differentiation and death, butyrate treatment did not affect the normal proliferation and regeneration of the injured epithelium in healthy cells, differentiated cultures, or in vivo experiments [159]. This phenomenon was dubbed “the butyrate paradox” [160,161,162]. Later, a possible explanation was suggested by the disclosure of the butyrate molecular mechanism which comprises the following: (a) activation of the G protein-coupled receptor 109a (GPR109a)–AKT signalling pathway, which leads to the remarkable inhibition of glucose metabolism and DNA synthesis in CRC cells, via reducing the amount of membrane G6PD and GLUT1 [163]; (b) the inhibition of AKT/ERK signalling in a histone deacetylase (HDAC)-dependent manner [164]. In malignant colonocytes, where glycolytic metabolism prevails over oxidative phosphorylation, butyrate accumulates and functions as an HDAC inhibitor, slowing the cell cycle progression through altered gene expression [165]. Thus, distinct metabolic pathways for cellular energy in differentiated and undifferentiated colonocytes are likely to be responsible for ‘the butyrate paradox’ [166]. During the coevolution of the microbiota with its hosts, mammalian crypt architecture has been developed to protect stem/progenitor cell proliferation from the potentially harmful effect of microbially derived butyrate; differentiated colonocytes establish a metabolic barrier that uses butyrate to produce a butyrate gradient [167]. Interestingly, butyrate, but not propionate or acetate, had a significant inhibitory effect on stem cell proliferation. This may be the reason why colonocytes, to protect intestinal stem cells, preferentially break down butyrate over the other SCFAs propionate and acetate, which are also present in high concentrations in the colon [167]. According to the mentioned theories, Lee et al. found out that metformin-butyrate (MFB), a new metformin derivative, showed more effective targeting of the CD44^+/high^/CD24^−/low^ CSC-like (undifferentiated) population in breast cancer in vitro and in vivo and the inhibition of mammosphere formation, compared to that with metformin [168]. Of note, when butyrate and 5-FU were administered together, the chemotherapeutic effectiveness of 5-FU on CRC cells increased, suggesting a role of butyrate in sensitising CRC cells to chemotherapy [163]. Moreover, in 3D-cultured organoids derived from CRC patients, when compared to that with the administration of radiation alone, butyrate dramatically increased radiation’s ability to cause cell death and improve therapeutic effects [169].

### 5.2. Secondary Biliary Acids

Dietary fatty acids may increase the ability of intestinal stem cells and progenitor cells to self-renew, as well as their capability to initiate tumours [170]. Bile acids are crucial signalling molecules that aid in the digestion and absorption of dietary lipids by acting as emulsifiers [171]. Cholic acid and chenodeoxycholic acid, the two primary biliary acids (BAs), are produced from cholesterol via a series of enzymatic processes that occur mostly in the liver. After being synthesised, these BAs are conjugated with glycine or taurine and subsequently secreted and stored in the gallbladder. Less than 5% of the BA pool enters the colon each day in humans due to an active transport mechanism that predominantly recycles BAs in the terminal ileum. The gastrointestinal microbiota metabolises BAs that enter the colon, converting primary BAs into secondary BAs, deoxycholic acid (DOC or DCA), and lithocholic acid (LCA). Hence, the circulating BA pool comprises approximately 30 to 40% of cholic acid and chenodeoxycholic acid, 20 to 30% of DOC, and less than 5% of LCA (in the conjugated form when it leaves the gallbladder and subsequently de-conjugated after it enters the colon via the action of bacterial enzymes) [172]. Secondary BAs are potent signal molecules that regulate a variety of processes (both physiological and pathological), through the modulation of several signalling pathways. Gut dysbiosis can alter the homeostatic levels of primary and secondary bile acid pools and produce distinct pathophysiological bile acid profiles [173]. Moreover, the gut microbiota–bile acid axis can control immune cells to indirectly promote tumours. Secondary BAs can inhibit the function of anti-tumour immune cells, such as macrophages, dendritic cells, B cells, and natural killer (NK) cells, while enhancing the function of Tregs, which are known to encourage the development of immunosuppressive microenvironments and the growth of tumours [174]. According to Bayerdorffer et al., there is a positive association between the colon-derived unconjugated fraction of DCA and colorectal adenoma formation, which are the precursors of CRC. The finding of this connection provided evidence in favour of the theory that DCA has a detrimental impact on colon cancer development [175]. Later, the mechanisms through which secondary BAs control carcinogenesis were described by Farhana et al. [176]. They discovered that the unconjugated secondary bile acids, notably DCA and LCA, alter muscarinic acetylcholine receptor M3 (M3R) and Wnt/β-catenin signalling promoting cancer stemness in colonic epithelial cells. Moreover, according to another study, secondary BAs can encourage the development of CSCs from both cancer and non-cancerous cells [174]. Farnesoid X receptor (FXR) is the nuclear receptor responsible for the negative feedback control of bile acid synthesis in the ileum and liver. Besides this role, FXR is also a crucial regulator of the proliferation of intestinal stem cells. In 2019, Fu et al. demonstrated that DCA and tauro-β-muricholic acid (T-βMCA) antagonise intestinal FXR, functioning as strong promoters of CSC proliferation able to induce DNA damage [177]. In their study, the authors also suggest that FXR activation could potentially impede tumour progression. They used the FXR agonist drug Fexaramine D to prove their theory, showing that when intestinal FXR is specifically activated, adenomas and adenocarcinomas in treated mice develop more slowly. A few years earlier, another research group identified two bacterial strains capable of directly modulating the activation of intestinal FXR [178]. They demonstrated that *Bacteroides dorei* and *Eubacterium limosum* cell-free supernatants trigger FXR activity and the expression of FXR-dependent genes in in vitro cell-based reporter assays and diet-induced obese (DIO) mice. Taken together, these results suggest that those two bacterial strains could have a beneficial role as probiotics, especially in those cases in which the (high-fat) diet is responsible for an imbalance in the BA pool that could favour CRC onset. A recent report suggests that in the presence of metastatic lesions, a healthy diet and/or proper pharmacological intervention aimed at re-establishing physiological bile acid levels could reduce cancer cell invasion, migration, and adhesion [173].

### 5.3. Cadaverine and Indolepropionic Acid

Lysine decarboxylase (LDC), a peculiar microbial enzyme, catalyses the decarboxylation of lysine to produce the bacterial metabolite cadaverine. Although cadaverine can also be produced by human cells, it appears that bacterial cadaverine production predominates over human biosynthesis [179]. Kovács et al. administered cadaverine in breast cancer cell lines within the standard range for serum (100–800 nM) and found that cadaverine exposure prevented mesenchymal-to-epithelial transition, inhibited invasion, and decreased mitochondrial oxidation, all hallmarks of stemness. Moreover, smaller and lower-grade primary tumours, together with reduced metastasis, were generated in Balb/c female mice transplanted with 4T1 breast cancer cells and treated with cadaverine [179].

L-tryptophan (Trp) is one of the nine essential amino acids for humans, and therefore, it must be introduced with the diet. Trp and other amino acids are released from dietary and endogenous luminal protein by bacterial proteases and peptidases. Three rate-limiting enzymes convert the Trp into kynurenine (Kyn): liver tryptophan-2,3-dioxygenase (TDO) and peripheral tissue indoleamine 2,3-dioxygenase 1/2 (IDO1/IDO2) [180]. Through the action of the bacterial enzyme tryptophanase, the intestinal microbiota mostly converts Trp into indole [181]. For human health, Trp metabolism through the Kyn pathway and gut microbial metabolism to indolic compounds is essential. For instance, breast cancer and breast cancer survival are strongly correlated with Trp and indole metabolism. In this regard, tumour TDO/IDO overexpression is a marker of poor prognosis [182,183]. Indeed, patients with breast cancer benefit from indole derivatives in terms of survival; of note, the levels of indole derivatives decrease with disease progression [184]. Reduced activity of the indolic pathway was seen in colon cancer, which also exhibits alterations in microbial indole synthesis [185]. As per Kovács et al., Sári and colleagues also employed the Aldefluor Stem Cell kit to measure the impact of treatment with indolepropionic acid (IPA), a bacterial Trp metabolite, on ALDH activity in 4T1 cells. What they discovered was a reduction in the percentage of aldehyde dehydrogenase-positive cells together with induced mesenchymal-to-epithelial transition (MET) in IPA-treated cells [184].

### 5.4. Ellagic Acid and Urolithins

The health-promoting potential of plant extracts and plant-derived secondary metabolites is widely recognised [186,187,188]. Numerous beneficial effects of polyphenols on human health, such as antioxidant [189,190,191,192,193], anti-inflammatory [194,195,196], immunomodulatory [197,198,199], cardioprotective [200,201,202], neuroprotective [203,204,205], anti-carcinogenic [206,207,208], and prebiotic properties [209], have been reported. Thanks to the plethora of chemical structures they exhibit, natural anticancer compounds may act as cytotoxic agents [210,211,212], anti-mitotic agents [213], angiogenesis inhibitors [214,215], topoisomerase inhibitors [216], apoptosis inducers [217] and cancer invasion [218], migration [219] and proliferation inhibitors [220,221,222]. The identification of plant-derived secondary metabolites that could target CSCs’ peculiar signalling has received much attention in current anticancer drug discovery approaches [223,224,225,226,227,228,229,230,231,232,233,234,235,236,237]. Very recently, a growing understanding of the impact of secondary polyphenol metabolites derived from gut microbial metabolism in the context of carcinogenesis has emerged. It is worth noting that the portion of dietary polyphenol that is absorbed at the small intestine level and enters the blood circulation is estimated at around 10%. Hence, many ingested polyphenols reach the large intestine, where intestinal bacteria convert them to phenolic acids [238].

*Lactobacillus rhamnosus*, an obligatory anaerobic homofermentative lactic acid producer, has been identified as predominant bacteria in the human gut [239]. The fermentation of polyphenol-rich dried black chokeberry (*Aronia melanocarpa*) powder using *L. rhamnosus* led Choi et al. to the isolation of a CSC inhibitor of which the structure was established as 1,2-dihydroxybenzene, also known as catechol [240]. In particular, they found that catechol inhibits proliferation and mammosphere formation in the human breast cancer cell lines MCF-7 and MDA-MB-231. Moreover, the percentage of breast cancer cells expressing CD44^high^/CD24^low^, as well as the protein and transcript levels of signal transducer and activator of transcription 3 (STAT3) and IL-6, are reduced by catechol treatment. Finally, catechol was found to reduce the expression of self-renewal genes, such as *NANOG*, *SOX2*, and *OCT4*, in CSCs, hence reducing their stemness and proliferative capacity.

Urolithins are secondary polyphenol metabolites generated via the activity of gut bacteria on ellagitannins (ET) and ellagic acid-rich foods, such as pomegranates, raspberries, strawberries, and walnuts [241]. The acid hydrolysis of ellagitannins releases free ellagic acid [242], which is further processed by gut microbiota that converts ellagic acid into urolithins [238]. The composition of a person’s gut microbiota affects how ellagitannins and ellagic acid are metabolised into urolithins; accordingly, individuals can be categorised into three groups of polyphenol-metabolising phenotypes called metabotypes [243]. Núñez-Sánchez and colleagues evaluated the effects of mixed ET-derived colonic metabolites on colon CSC-associated markers [244]. The authors investigated the ability of two separate mixtures of compounds—ET metabolites, ellagic acid (EA), and the gut microbiota-derived urolithins (Uro)—that, in proportion and concentration, mimic those detected in vivo in individuals with metabotype-A or metabotype-B. According to their study, the mixture resembling the metabotype-A that contains mostly Uro-A (85% Uro-A, 10% Uro-C, 5% EA) was more successful at suppressing CSCs’ molecular (ALDH activity) and phenotypic (number and size of colonospheres) traits, whereas the mixture mimicking the metabotype-B containing less Uro-A but IsoUro-A and Uro-B (30% Uro-A, 50% IsoUro-A, 10% Uro-B, 5% Uro-C, 5% EA) seemed to have some effects on colonosphere size and number, but not on ALDH activity levels. Uro-A, the predominant metabolite in the metabotype-A mixture, may be the main factor causing the discrepancies seen between the two mixtures. Interestingly, González-Sarrías et al. also reported that Uro-A is a substrate of drug efflux transporter breast cancer resistance protein (ABCG2/BCRP), highlighting the role of Uro-A in targeting CSCs [245]. In addition, the finding that the anticancer activity of 5-FU can be enhanced by Uro-A in human colon cancer cells supports the hypothesis that using phytochemicals in combination with traditional cytotoxic drugs to target CSCs may be a new cancer treatment approach [246].

### 5.5. Retinoids

Diet is the primary source of vitamin A since it cannot be synthesised by animal tissue and has to be introduced with food. Retinoids (including vitamin A, all-trans retinoic acid, and related signalling molecules) were shown to promote the differentiation of diverse stem cell types [247]. Retinoic acid (RA), a well-known vitamin A metabolite, regulates the fate of neighbouring cells. The availability of vitamin A (retinol), the activity of the enzymes necessary for RA synthesis (retinol dehydrogenases and aldehyde dehydrogenases), and the catabolism of RA by CYP26 enzymes all affect the levels of RA [248]. Retinoid signalling is frequently impaired early in carcinogenesis, suggesting that a decrease in retinoid signalling may be essential for tumour growth [249]. Although RA has frequently been regarded as a cell differentiation inducer, depending on the type of cell, RA might prevent cell differentiation and induce stemness [248]. Recent discoveries of retinoids as chemo-preventive and molecular-targeted antitumour agents reveal that RA agents may be considered efficient therapies for treating human solid tumours [250]. Among retinoids, all-trans retinoic acid (atRA) was found to be a promising therapeutic compound capable of targeting CSCs in different cancer settings, such as gastric [251], brain [252], head and neck [253], and breast [254] cancer. For instance, a significantly improved anti-cancer effect towards breast cancer was achieved when atRA and DOX were simultaneously delivered, encapsulated in the same nanoparticle [255]. This combinational drug delivery system aims to target both non-CSCs and CSCs. With their studies in vitro and in vivo, Sun et al. demonstrated that the atRA-induced differentiation of CSCs into non-CSCs can decrease their capacity for self-renewal and enhance their sensitivity to DOX, improving the inhibition of tumour growth while simultaneously decreasing the incidence of CSCs. Moreover, in A549GSC and H1650GSC cells, treatment with atRA was shown to dramatically lower the IC50 values for gefitinib, an ATP-competitive EGFR tyrosine kinase inhibitor used in non-small cells lung cancer (NSCLC) treatment, and the high expression of ALDH 1 family member A1 (ALDH1A1) and CD44 [256]. Additionally, conventional PKC inhibitor (Gö6976) and atRA combined treatment reduced tumour growth, metastatic dissemination, and the frequency of breast CSCs in vivo while impairing the proliferation, self-renewal, and clonogenicity ability of breast CSCs [257]. Interestingly, both products and substrates of the RA pathway, 5 μM atRA and 1 μM ROL, respectively, were shown to inhibit ALDH1^+^ CSC populations in cisplatin-resistant NSCLC cells [258]. Recently, Bonakdar et al. showed the importance of gut bacteria and their ability to metabolise vitamin A to produce a variety of retinoids with pharmacological activity [259]. In particular, they compared the retinoid metabolomes from caecal contents from germ-free (GF), conventional (CV), and antibiotic-treated mice (CV + Abx) and demonstrated that (1) GF mice had notably reduced amounts of all-trans-retinol (atROL), atRA, and 13-cis-retinoic acid (13cisRA) compared with those in CV mice and (2) when compared to that in control mice, CV animals treated with an antibiotic cocktail displayed a marked decrease in concentrations of all vitamin A metabolites except for RE. These results indicate that dietary vitamin A can be converted into ROL and its active metabolites, atRA and 13cisRA, by the gut microbiota. Besides the above-mentioned anticancer potential of atRA, it is worth noting that 13cisRA, also known as isotretinoin, is a key treatment for treating high-risk neuroblastoma and for dermatology. The presence of 13cisRA in the mouse caecum of CV mice but not GF or CV + Abx mice, as well as its in vitro production by caecal bacteria, indicates that 13cisRA is a particular retinoid derived exclusively from microorganism metabolism [259].

## 6. Conclusions

The identification of CSCs as a significant contributor to and driver of cancer development mechanisms, such as tumour growth, recurrence, metastasis, and therapy resistance, constitutes a significant advancement in the study of cancer and offers researchers the opportunity to develop new CSC-centric approaches for cancer treatment. The failure of cancer therapy is mostly due to CSC cell-mediated drug resistance. Characterising the differences between non-neoplastic tissue stem-cell programs and those of neoplastic tissue stem cells will be critical in developing therapeutic strategies to selectively target CSCs without negatively affecting non-neoplastic tissue stem cells. The development of mechanism-based methods for cancer drug discovery, including targeted therapies and immunotherapies, has been aided by remarkable improvements in our understanding of the molecular basis of cancer and tumour cell biology. However, there is a pressing need for the development of therapeutic approaches that are more successful in overcoming CSC cell-mediated resistance. In this regard, efforts are currently being made to find effective, affordable, and safe anticancer medicines of natural origin. There are now several strong connections among the host’s nutrition, the composition of the gut microbiota, and the host’s physiology. Particularly, numerous reports have underlined the key role of diet in cancer prevention [260,261]. For instance, it has been proven that the Mediterranean diet regimen significantly lowers the risk of several cancers, particularly colorectal and aerodigestive [262,263,264], gastric [265], pancreatic [266], breast [267,268,269], nasopharyngeal [270], lung [271], prostate [272], and bladder cancer [273].

The impact of the human microbiota on both short- and long-term human health has been amply shown during the last few decades [274,275,276,277,278]. In recent years, growing evidence has indicated the causal relationship between intestinal microbial dysbiosis and colorectal cancer aetiology [279]. In this perspective, to reverse established microbial dysbiosis, a range of approaches has been employed, including probiotics, prebiotics, postbiotics, antibiotics, and faecal microbiota transplantation (FMT) [280]. Currently, the small molecular weight compounds (postbiotics) released by the microbiota, which provide the host with many physiological health benefits, are given much attention. The host’s biochemical versatility is increased by the large metabolic repertoire of the microbial population, which supports the activity of mammalian enzymes and allows the host to metabolise a variety of food substrates [281]. This diet–microbial metabolism feedforward loop modulates a broad spectrum of events. Here, we reviewed the emerging roles of microbiota-derived metabolites as CSC-targeting anticancer agents. The body of evidence provided suggests that postbiotics, bioactive substances derived from gut beneficial microbiota, might be considered novel promising agents to be used in personalised medicine approaches to re-establish gut eubiosis while also targeting CSCs. This strategy may encompass the steering of diet–microbiota interactions toward the production of certain metabolites that could maximise health benefits. Furthermore, the synergistic effect of diverse microbial products with standard anticancer agents may suggest their further employment to sensitise CSCs in chemo-/radiotherapy regimens. In this perspective, postbiotics are superior to probiotics for industrial production because they are easier to use and store, have a longer shelf life, are stable across a wide pH and temperature range, and do not produce bioamine. However, before postbiotics can be employed as probiotic substitutes, more investigation is needed into the production, distribution mechanisms, and safety standards of medicines and functional foods [137]. Moreover, a crucial aspect to take into account from the viewpoint of postbiotic-based therapeutics is their targeted delivery in vivo. Indeed, it is crucial to ensure that a biomolecule given orally, intravenously, or topically can be transferred to its site of action without being altered via pharmacokinetics or digestive processes. In this regard, a recent summary of possible methods for the in vivo delivery of postbiotics was provided by Abbassi et al. [131]. In conclusion, the reviewed literature highlights that the microbiota is a valuable resource for the discovery of novel small-molecule drugs, and metabolites originating from the microbiota may find extensive use in the treatment of cancer, thanks to their ability to target CSCs. In this respect, further study of the pharmacological interaction between conventional chemotherapeutic drugs and gut microbiota-derived compounds will undoubtedly be necessary for the development of improved therapeutic approaches to eliminate CSCs.

## 7. Data Collection

For the current review, data were gathered from English-language scientific publications using different combinations of the following keywords: ‘cancer stem cells’, ‘cancer’, ‘stemness’, ‘signalling pathway’, ‘microbial products’, ‘microbiota metabolites’, ‘bacterial products’, ‘bacterial metabolites’, ‘probiotic ghosts’, ‘postbiotics’ as keywords in search queries of different databases and electronic search engines. Publications addressing CSC-associated mechanisms of therapeutic resistance, and articles describing the activity of gut microbiota bioactive metabolites toward CSC features were selected.

## Figures and Tables

**Figure 1 ijms-24-04997-f001:**
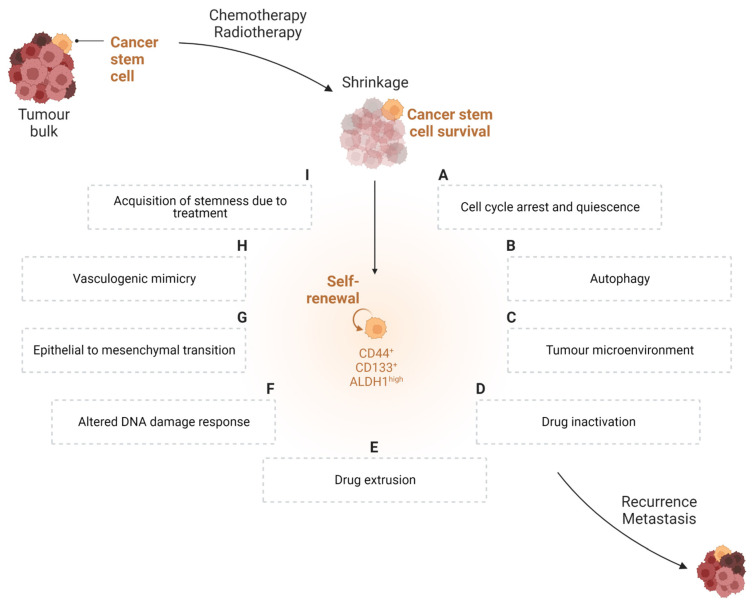
**Schematic representation of the different mechanisms applied by CSCs to escape cancer therapy.** A small number of cancer cells, known as cancer stem cells (CSCs), have a significant role in the failure of cancer treatment. Despite chemotherapy successfully eliminating a significant amount of the tumour bulk, the main factor for tumour recurrence and metastasis is the existence of CSCs that are resistant to chemotherapy and can regenerate themselves. CSC-mediated therapy resistance appears to be attributed to different mechanisms: cell cycle arrest and quiescence (**A**), autophagy (**B**), interactions with the tumour microenvironment (**C**), drug inactivation (**D**) and extrusion (**E**), alteration of the DNA damage response (**F**), epithelial-to-mesenchymal transition (**G**), and vasculogenic mimicry (**H**). Moreover, stemness-related therapy resistance could be induced by cancer treatment itself (**I**).

**Figure 2 ijms-24-04997-f002:**
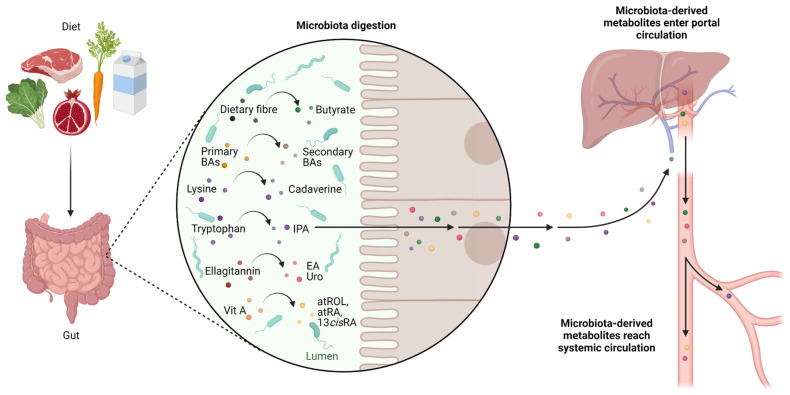
**Schematic representation of the metabolites produced via microbiota digestion of dietary compounds that have the potential to target CSCs**. The gut microbiota plays a role in digestion by metabolising indigestible macronutrients. The host’s metabolic capability is increased by the large enzymatic repertoire of the microbial population, which integrates the function of mammalian enzymes and allows the host to metabolise a variety of food substrates. Numerous bacterial metabolites are produced by the intestinal microbiota’s metabolic activities toward the available substrates and may accumulate in the lumen. Microbiota-derived metabolites possess enhanced or even different bioactivities compared to their parental compounds. Moreover, they can access circulation and potentially diffuse systemically. Specific products of microbial digestion, highlighted in the zoomed callout, have been found to target CSC features. BAs, biliary acids; IPA, indolepropionic acid; EA, ellagic acid; Uro, urolithins; atROL, all-trans-retinol; atRA, all-trans retinoic acid; 13cisRA, 13-cis-retinoic acid.

## Data Availability

No new data were created. Data sharing is not applicable to this review.

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
