# Peer review of "Microbiota-Derived Natural Products Targeting Cancer Stem Cells: Inside the Gut Pharma Factory"

_ijms, 2023, doi:10.3390/ijms24054997_

Round 1

Reviewer 1 Report

1.     Although the authors pointed out two aspects to be reviewed. As for the first aspect (Part 3), most of the contents described the key process of tumor progression, without further refining the process related to CSCs therapy resistance. The authors need to make major revision of this part.

2.     In line 362-377, the authors mentioned the small molecules and postbiotics from microbiota metabolic on the TME and their roles in cancer development. The authors are advised to summarize these reported small molecules and products in Tables, were assays in vitro, in vivo, preclinical or clinical trials?

3.     In section 5.5 Vitamins, the author only mentioned the role of vitamin A and retinoids in the treatment of CSCs, but did not mention the role of other kinds of vitamins in the treatment of CSCs at all. Is there no research? Or did the authors not summarize it completely? The same problems also existed in the other section of Part 5. The authors need to conduct sufficient literature to further improve and modify this part.

4.     Abbreviations that appear for the first time in the manuscript should be clearly written with their full names. And avoid alternating between full names and abbreviations throughout the manuscript. Please checked carefully and revised the manuscript.

Author Response

We would like to thank the reviewer for the detailed comments he/she provided in order to improve our manuscript. We addressed every suggestion point by point as detailed in the file attached.

Reviewer 2 Report

Artusa et al. present a comprehensive view of the link between the gut microbiome landscape and cancer stem cells regarding various perspectives such as drug resistance, recurrence, and other tumor hallmarks. However, this paper is timely and relevant in tumor biology for diagnostic and therapeutics. However, some points are suggested that can enhance the impact of this paper.

1. Since the title of the paper is microbiota-derived Natural Products, therefore a major emphasis should be given to microbiotas derived chemicals and cancer stem cells. In general, about CSCs and other mechanisms, several reviews are already published.

2. A discussion on link between CSCs, gut microbiotas and cancer cachexia may be explored.

3. Some evidence of preclinical and clinical breakthroughs in gut-derived products and anticancer approaches can be explored.

4. additional views on gut microbiota-derived products and non-human models and their implications in CSC-mediated tumor hallmarks can be explored.

5. The fond, typo errors, and image quality need to be improved. 

Author Response

(The authors gave the same response as above.)
